# Group-Based Pelvic Floor Telerehabilitation to Treat Urinary Incontinence in Older Women: A Feasibility Study

**DOI:** 10.3390/ijerph20105791

**Published:** 2023-05-11

**Authors:** Mélanie Le Berre, Johanne Filiatrault, Barbara Reichetzer, Chantale Dumoulin

**Affiliations:** 1School of Rehabilitation, Faculty of Medicine, Université de Montréal, Montreal, QC H3N 1X7, Canada; 2Research Center, the Institut Universitaire de Gériatrie de Montréal (CRIUGM), Montreal, QC H3W 1W4, Canada; 3Department of Obstetrics and Gynecology, Centre Hospitalier de l’Université de Montréal (CHUM), Montreal, QC H2X 0C1, Canada; 4Department of Obstetrics and Gynecology, Faculty of Medicine, Université de Montréal, Montreal, QC H3C 3J7, Canada; 5Institut Universitaire de Gériatrie de Montréal (IUGM), Montreal, QC H3W 1W5, Canada

**Keywords:** urinary incontinence, telerehabilitation, aged, feasibility studies, women’s health

## Abstract

Less than half of women with urinary incontinence (UI) receive treatment, despite the high prevalence and negative impact of UI and the evidence supporting the efficacy of pelvic floor muscle training (PFMT). A non-inferiority randomized controlled trial aiming to support healthcare systems in delivering continence care showed that group-based PFMT was non-inferior and more cost-effective than individual PFMT to treat UI in older women. Recently, the COVID-19 pandemic highlighted the importance of providing online treatment options. Therefore, this pilot study aimed to assess the feasibility of an online group-based PFMT program for UI in older women. Thirty-four older women took part in the program. Feasibility was assessed from both participant and clinician perspectives. One woman dropped out. Participants attended 95.2% of all scheduled sessions, and the majority (32/33, 97.0%) completed their home exercises 4 to 5 times per week. Most women (71.9%) were completely satisfied with the program’s effects on their UI symptoms after completion. Only 3 women (9.1%) reported that they would like to receive additional treatment. Physiotherapists reported high acceptability. The fidelity to the original program guidelines was also good. An online group-based PFMT program appears feasible for the treatment of UI in older women, from both participant and clinician perspectives.

## 1. Introduction

Urinary incontinence (UI) is one of the most prevalent health concerns in women age 65 and over [1,2], with half of community-dwelling older women suffering from UI [2]. Left untreated, UI can have tremendous consequences on the overall health and quality of life of older women [3,4]. As a recognized risk factor for institutionalization [5], and a costly condition, both for women themselves [6] and for the healthcare systems, UI definitely stands out as an important health condition that needs to be addressed in a timely manner. The recommended first-line treatment for UI in women is individual pelvic floor muscle training (PFMT) (Level of evidence 1; Grade of recommendation A) [7]. Despite this strong recommendation, more than half of women with UI are not receiving treatment [8,9,10]. Hence, healthcare systems worldwide currently are in need of further solutions to meet the needs of women [11]. Treatment accessibility still appears hampered by lack of human resources and financial constraints [12]. As a response to these needs, a recent non-inferiority randomized controlled trial—the Group Rehabilitation Or IndividUal Physiotherapy (GROUP) trial (ClinicalTrials.gov, NCT02039830)—showed that group-based PFMT was not inferior to individual PFMT for treating UI in older women, despite using fewer resources and, thus, being more cost-effective [13,14,15,16]. Other studies have also shown the benefits of group-based PFMT on UI symptoms in adults [17] and older women [18,19,20,21]. Group-based PFMT has now gained official clinical recognition and has a grade B recommendation for the treatment of UI in postmenopausal women in the latest version of the International Consultation on Incontinence (ICI) reference book [22].

However, the COVID-19 pandemic prevented group gatherings, especially for older adults, who were at higher risk of complications. For the past two years, it therefore limited the implementation of in-person group-based interventions, such as the GROUP trial’s program. At the same time, the pandemic also drove an impressive surge in many remotely delivered healthcare services [23], which highlights the need to provide online options for group-based PFMT.

One promising solution is telerehabilitation, which refers here to the remote delivery of synchronous rehabilitation services using information and communication technology. It has already shown good feasibility and favorable clinical outcomes for the treatment of various orthopedic and neurological conditions [24,25,26,27,28,29,30,31,32,33]. Recently, individual PFMT telerehabilitation also showed comparable effectiveness to individual in-person PFMT for the treatment of UI [34]. However, no study to date has investigated the feasibility of delivering group-based PFMT telerehabilitation. This approach holds the potential to enhance the accessibility of continence care during health crises, particularly for women living in rural or remote areas, or with mobility and transportation issues.

This study thus aimed to assess the feasibility of an online adaptation of the GROUP program (the teleGROUP program) for UI in women age 65 and over, from both participant and clinician perspectives. Feasibility studies are a critical component of the evaluation of study processes. They establish the necessary groundwork to shape the subsequent phases of clinical development and provide insights for successful implementation [35,36,37,38]. Feasibility constitutes in itself a valid and relevant contribution to clinical evidence on a given treatment approach, alongside appropriateness and effectiveness [39], and holds the potential to bridge the gap between research and practice [38]. Therefore, assessing the feasibility of this online group-based PFMT program in this study represents the very first step in further developing this innovative UI treatment option for older women.

## 2. Materials and Methods

### 2.1. Study Design

This pilot pre-post feasibility study is part of a larger research program aiming to assess the feasibility, acceptability, effects and costs of the teleGROUP program (ClinicalTrials.gov NCT05182632). The present study focused on feasibility. More specifically, the study aimed to determine the feasibility of teleGROUP in terms of attendance, adherence, complications, dropouts, satisfaction and the overall experience data of participating women; acceptability for the evaluating physiotherapists; and conformity with the original program guidelines from the physiotherapist leading the teleGROUP program. This study follows the Consolidated Standards of Reporting Trials (CONSORT) for reporting results. A detailed study protocol was previously published [40].

### 2.2. Participants

The research team recruited participants through advertisements, a research participant database and referrals from collaborating clinics. Women were eligible if they were age 65 and older; able to walk independently; presented stress or mixed UI, as confirmed by the Questionnaire for Incontinence Diagnosis (QUID) [41], with at least three weekly urine leakages, persisting for three months or more [14,15]; had internet access; and presented no cognitive deficit (Mini Mental State Examination (MMSE) score of 24/30 or more) [42]. Women were not eligible if they reported any condition that could interfere with PFMT or the study processes. The complete list of eligibility criteria is available in previously published protocols [40,43].

### 2.3. Intervention

During an initial individual in-person evaluation session, participant eligibility was confirmed by an experienced local physiotherapist with specialized training in pelvic floor rehabilitation. During this session, the physiotherapist also took the time to teach the participant how to correctly contract her pelvic floor muscles (PFMs) through digital vaginal palpation and verbal cues [44].

Participants who were able to voluntarily contract their pelvic floor muscles following a verbal command by the end of the evaluation session were deemed eligible and then took part in the teleGROUP program [14,15]. TeleGROUP is an online group-based PFMT program comprising 12 weekly 1-hour training sessions. A thirteenth optional session was offered to all participants at the end of the program to compensate for any missed sessions. An experienced pelvic floor physiotherapist delivered all sessions via Zoom. In accordance with the initial GROUP trial protocol [14,15], we aimed to form groups of six to eight women. If needed, participating women received an introduction to Zoom and support over the phone one week before the start of the program. The local physiotherapist conducting the evaluation session also provided an exercise booklet and educational support material to each participant.

A videoconference connection link, which was unique for each cohort, was sent weekly via email to all participants two to three days before the session. No login was required, and participants could click to join the session directly from the email they received. Each weekly virtual session began with a 1–3 min individual exchange between the physiotherapist and each participating woman in a private breakout room to quantify UI episodes and exercise adherence in the previous week. In the meantime, the rest of the group were invited to socialize in the Zoom meeting’s “main room”. All sessions were then divided into a 10–15 min educational component and a 30–45 min PFMT exercise component. More detail on both the educational and exercise components of the teleGROUP program are available in Appendix A and in previously published protocols [15,40]. In the case of an unexplained absence for a weekly session, the research team ensured that the participant was subsequently contacted via phone or email for a follow-up. In addition to their weekly group session, participating women were also invited to complete a home exercise program including four PFMT exercises, five days per week. The four exercises in this home exercise program targeted strength, speed of contraction, endurance and coordination, and progressed over three phases lasting four weeks each, allowing for increasing difficulty in the duration, number of repetitions and position (from lying, to sitting, to standing), in line with the weekly virtual treatment sessions [14]. Upon program completion, participants were introduced to a six-month maintenance exercise regimen.

### 2.4. Data Collection

The research team pre-screened women for eligibility through a telephone interview. A specialized physiotherapist located in the participant’s region then confirmed their eligibility through an individual in-person evaluation. During this evaluation, the physiotherapist collected sociodemographic and health data, namely, age, height, weight, socioeconomic status, medical history and general health characteristics, including type of UI symptoms, duration of symptoms and cognitive status through the MMSE [42] and Montreal Cognitive Assessment (MoCA) [45,46]. Participants completed a 7-day bladder diary prior to their initial individual in-person evaluation session [47,48,49]. They also completed the International Consultation on Incontinence Questionnaire module on UI symptoms (ICIQ-UI) short form [50], one question from the Broome Pelvic Muscle Exercise Self-Efficacy Scale (PMSES) [51] and the Online Technologies Self-efficacy Scale (OTSES) [52]. The physiotherapist also performed an intra-vaginal evaluation to assess PFM function through digital vaginal palpation [53,54]. 

To assess feasibility from the participant perspective, the physiotherapist leading the teleGROUP program recorded attendance at weekly sessions [55] and adherence to weekly home exercises [55], in addition to any complications or side effects reported. The research team also recorded dropout rates and reasons for dropping out. Additionally, at the end of the program, participants rated their satisfaction with the program’s effects on their UI symptoms using a single-item question (“completely satisfied”, “somewhat satisfied” or “not at all satisfied”) [56] and a visual analog scale, and described their experience with the program using the System Usability Scale (SUS) [57]. 

To assess feasibility from the clinician perspective, both the evaluating physiotherapists and the treating physiotherapist were considered. First, the physiotherapists conducting the initial individual in-person evaluation session completed a questionnaire based on the Theoretical Framework of Acceptability (TFA) [58], using similar questions to those used in the literature, adapted to the context of the study [59,60]. This anonymous online questionnaire was sent in June 2022 by email to all the physiotherapists, who conducted at least one initial individual in-person evaluation session with a participant. Up to three email reminders were sent to fill out the online questionnaire.

Secondly, still within the assessment of feasibility, the physiotherapist leading the teleGROUP program recorded fidelity to the original program guidelines at each session for the full 12 weeks [55] using a self-report conformity checklist of the GROUP trial’s program content [61]. A physiotherapy doctoral student observed four sessions and independently completed the same conformity checklist to ensure the validity of the self-report checklist answers.

### 2.5. Data Analysis

Data were tabulated and interpreted using descriptive statistics, such as means, medians, standard deviations and interquartile ranges for continuous variables and frequency distributions for categorical and dichotomous variables, as appropriate.

## 3. Results

Participants were recruited from March 2021 to April 2022. Of the 150 women who were interested in participating in the study, most were contacted through community groups, associations or public events (51/150, 34%), journals or magazines (32/150, 21%), a participant database (22/150, 15%) or through word of mouth and past experiences with the research team (16/150, 11%) (Appendix A). Among them, 99 (66%) did not meet the study’s inclusion criteria. In total, 28 (28.7%) had other types of UI symptoms than stress or mixed UI, 24 (16%) had mild symptoms only, and 47 (31%) reported a clinical profile incompatible with the study, such as chronic constipation [62] or currently taking UI medication. Among them, 2 were unable to perform a voluntary PFM contraction following verbal command (1%), even with the help of vaginal digital palpation and verbal cues. Additionally, 10 women (15%), all of whom were contacted from a general research participant database, declined participation after discussing the details of the study involvement. All ineligible women were referred to appropriate resources available in their region.

Thirty-four women were included in the study and divided into four cohorts. Of those, 33 completed the program. Eight participants requested the Zoom introduction and technical support over the phone before the program. Participants waited a median (IQR) of 21 (26) days between their initial recruitment phone contact and in-person evaluation. There was a median of 26 (33) days between their in-person evaluation and first session of the program. These wait times were due to the availability of the participants and physiotherapists, in addition to the time required to receive and complete the 7-day bladder diary prior to their initial individual in-person evaluation session. 1 participant (3%) dropped out after the seventh week of the program due to personal reasons, and 1 participant (3%) was unreachable and lost to follow-up after program completion, leading to an attrition rate of 2/34 (6%) for the study. The complete flowchart of participants is available in Figure 1. The median age of the initial sample of women was 69 years old. Most participants (94%) reported symptoms of mixed UI, with a median of 5.5 years of symptom duration. Overall, women reported a median of 13.5 weekly urine leakages, and a mean ICIQ-UI SF score of 12.5 (SD = 3.0) out of 21, indicating moderate to severe UI [63,64]. At the individual in-person evaluation session, the median participant confidence in their ability to contract their PFMs was 80.0 (IQR 20.0) out of 100, showing high self-efficacy [51]. Regarding their self-efficacy with online technology, participants reported a mean OTSES score of 60.8 (SD = 20.7) out of 120, with higher values indicating lower self-efficacy [52]. No OTSES norms for older adults are currently available. The ability to perform a PFM contraction with the physiotherapist’s guidance was part of the inclusion criteria for the study. However, most participants (30/34, 88%) initially displayed muscular compensation when attempting to perform a PFM contraction. They compensated with either their abdominal, gluteal or adductor muscles or their diaphragm, but corrected their movement after verbal cues. Table 1 summarizes the characteristics of participating women.

Throughout the 12 sessions provided to each of the 4 cohorts of older women, no technological problem arose that could have prevented their participation in the online program. Table 2 summarizes the data used to examine the feasibility of the teleGROUP program from the participant perspective. Participating women attended 95% of all scheduled sessions throughout the program. They took part in a median of 12.5 available sessions, and 17 (50%) participated in a thirteenth “optional session”, even if they had not missed any of the 12 scheduled sessions. Most participants (30/33, 91%) reported that they completed their home exercises 4 to 5 times per week (Table 2, Figure 2). Only 1 participant (3%) reported a minor side effect, lower back muscle soreness during the first weeks of training, which resolved rapidly. Women reported a median percentage of satisfaction with the program’s effects on their UI symptoms of 75% after program completion. Only 1 woman (3%) was not satisfied, as she felt her symptoms did not improve with the program. The women scored a median SUS of 93.8, corresponding to an A+ grade or “best imaginable” usability for the teleGROUP program [57,65]. In their SUS questionnaire, most women declared that the teleGROUP program’s online format was easy to use, and they felt confident participating in the program. They provided a median rating of 5 out of 5 for both these questions. Furthermore, they thought that other women in a similar situation would learn very quickly to navigate the teleGROUP program. They provided a median rating of 5 out of 5 regarding their agreement to this SUS item.

Table 3 and Table 4 present the data used to examine the feasibility of the teleGROUP program from the clinician perspective. In total, 14 out of 16 physiotherapists, who conducted initial individual in-person evaluations, filled out the online questionnaire. They reported high acceptability regarding their involvement in the program. They had a median score of 9.0 or more out of 10.0 on every item of the TFA questionnaire (Table 3). The lowest acceptability scores pertained to two items of the questionnaire: the burden associated with planning and conducting the initial evaluations, as well as the missed opportunities associated with conducting an evaluation for a program delivered outside of their physiotherapy clinic, or opportunity costs. For these 2 items, the lowest scores were 1.0 and 5.0, and the interquartile range lower brackets reached 7.8 and 8.0, respectively. Adding new patients to their already busy schedule, sometimes on short notice, was a challenge for 2/14 (14%) physiotherapists, which explains the burden score. In addition, 2/14 (14%) physiotherapists expressed slight worries about missed business opportunities, where they may have to refer potential clients to another physiotherapist outside of their clinic, which explains the opportunity costs score. 

Secondly, the physiotherapist leading the teleGROUP program delivered the program to groups of 6 to 11 women, thus exceeding the initial group size established in the protocol. The time dedicated to individual exchanges with the physiotherapist at the beginning of each session was longer than anticipated, which demonstrates the appreciation and unexpected enthusiasm of participants for this component of the program. Yet, it resulted in some subsequent time constraints. Time constraints increased with additional participants in the group, and program delivery took an additional 10 to 20 min, depending on the number of women present at the session and the time spent in individual exchanges. Throughout the program, fidelity to the program content was good, as reported by the leading physiotherapist, with only a few non-central elements that were not completed (Table 4). The physiotherapist’s self-reported conformity to the program content concurred with the doctoral student’s independent assessment of the conformity. Overall, the physiotherapist left out the non-PFM exercises of the exercise component from the original program, such as a warm-up and flexibility movements, deep breathing and core training, because of time constraints. Conversely, the physiotherapist prioritized the four main PFM exercises, completing them according to the protocol or with a position change (Appendix A). The physiotherapist also added additional sets of exercises on some occasions, bringing the structure of the weekly exercise sessions of the program closer to the home exercise program. Finally, the physiotherapist reduced the length of the functional exercise ‘dance activity’ at the end of the session by two thirds in almost every session.

## 4. Discussion

This pilot feasibility study is the first to investigate the feasibility of an online group-based PFMT program. The study showed that the teleGROUP program was feasible from the participant perspective, with high attendance at the online classes, high adherence to the home exercises, infrequent and minor side effects resolving quickly, low attrition throughout the 12-week program, high satisfaction with the program and good perceived usability of the online program. In addition, this study showed good feasibility from the clinician perspective, with high acceptability from the evaluating physiotherapists and relatively good fidelity to the original program guidelines from the physiotherapist leading the program, despite time constraints. 

Compared to the original in-person version of the program, teleGROUP showed a similar attendance rate (95% vs. 95.2% in GROUP and teleGROUP, respectively) and lower attrition throughout the program (7% vs. 3% during the program and 6% for the overall study) [15]. Exercise adherence was also high in both studies (86% vs. 91% of women completed their exercises more than 4 times per week in the GROUP and teleGROUP research projects, respectively) [13]. As this study is the first to investigate online group-based PFMT, it was not possible to compare it to other existing programs. However, a recent systematic review aimed at examining the practice of pelvic floor telerehabilitation identified four RCTs [66]. While three of the RCTs investigated asynchronous PFM training, the fourth study described an eight-week continence management program consisting of weekly educational talks [67]. Although not a group-based PFMT program per se, the educational sessions of this continence management program covered the topic of PFM exercises, and participating women were encouraged to complete the PFM exercises at home. A specialist nurse delivered the program remotely to a group of women, who were all gathered in a room in a community center. A research assistant was present with the women to support the videoconferencing and the overall session delivery. This hybrid online/in-person format allowed the healthcare professional to lead the sessions remotely, similarly to teleGROUP, although the participants still had to travel and meet in person. Attendance and attrition rates for this group-based videoconferencing program were 99% and 3%, respectively, which is similar to the rates obtained in the present study. These comparisons suggest that an online group-based program can achieve similar attendance, attrition and adherence as the in-person GROUP program [13], and similar attendance and attrition as an hybrid online/in-person continence management program [67]. More recently, another systematic review investigated remote rehabilitation methods for the delivery of PFMT [68]. and concluded that there was a dire need for more research on synchronous communication methods in the field. Indeed, of the eight included RCTs, six targeted the use of training devices, one study focused on the use of a mobile application and one assessed the use of a web-based asynchronous program. Therefore, none of the studies used videoconferencing to provide remote PFMT. While some studies included monitoring or follow-up evaluations via phone calls or hospital and clinic visits, there were no PFMT sessions and no PFM physiotherapy treatment sessions delivered through synchronous telerehabilitation. This present study’s findings on the feasibility of a group-based PFMT program, therefore, seeks to bridge this gap and initiate the conversation on synchronous pelvic floor telerehabilitation.

Throughout the program, the clinical expertise of the specialized physiotherapists played a key role in ensuring adequate treatment delivery, regardless of the online or group format. First, during the initial in-person evaluation, the physiotherapists taught participants to contract their PFMs. They were able to guide the participating women so that they could achieve an adequate PFM contraction by overcoming perineal inversion (i.e., straining and depressing the pelvic floor rather than executing the expected inward lift and squeeze), compensations (e.g., contraction of other muscles, such as abdominal muscles, gluteal muscles, adductors, diaphragm) or difficulty relaxing their PFM. These challenges are similar to those observed in other studies, where between 17% and 43% of women initially strained, pushed or used a Valsalva maneuver when attempting to contract their PFMs [69,70]. 

Additionally, when facing time constraints during the 12-week program, the physiotherapist leading the teleGROUP sessions made some decisions that were informed by clinical expertise and experience. This clinical decision process was reflected in the fidelity findings: Besides the individual exchange and the educational session, the physiotherapist prioritized the four main PFM exercises constituting the core elements of the program over the other exercises from the program. Comparatively, in the GROUP trial, the per protocol fidelity percentage was higher overall (77.9% of the content for the functional exercise of the dance activity and 79.8% of the content for the transverse abdominal contractions (unpublished data), compared to 31.2% and 12.5%, respectively, in the teleGROUP program). Yet, the disparities between the in-person and online versions of the program tend to disappear when considering the core elements of the program, notably, the four main PFM exercises. Indeed, the fidelity was relatively similar for these 4 PFM exercises, with 90.7%, 96.2%, 100.0% and 98.7% for each exercise (unpublished data), compared to 96.7%, 97.0%, 98.4% and 100.0% in the GROUP and teleGROUP programs, respectively. Moreover, the physiotherapist leading the teleGROUP program limited the position changes during the sessions to avoid multiple camera angle adjustments by both the physiotherapist and the participating women and to reduce the time spent correcting posture. Additional time was still needed to complete the treatment sessions due to the time spent in individual exchanges, particularly when dealing with a high number of participants. As adherence to the PFM exercise program constitutes the cornerstone of successful UI treatment [71,72], the emphasis put on the four main PFM exercises by the physiotherapist supports the clinical objectives of reducing UI symptoms. Through this anchoring to specialized physiotherapy expertise, the program thus aims to remain focused on achieving its purpose. It aspires to combine and optimize the advantages of education, exercise and a group format to achieve the best possible outcomes for older women with UI.

The findings of the present study represent an important first step in guiding the upcoming research steps towards the implementation of group-based PFMT telerehabilitation programs in various healthcare settings. This format provides a safe UI treatment option during pandemic periods or in any other situation that presents a health risk (e.g., winter storm). An online option could also increase the accessibility of continence care for women living in rural or remote areas, where pelvic floor rehabilitation services are unavailable or scarce. It could also be an additional asset for any woman who cannot attend in-person treatment due to her living situation (i.e., caring for someone at home), transportation difficulties or any other personal reason. 

This study also presents some limitations. As this was a feasibility study on an innovative treatment approach, a relatively small sample size was included, and the intervention was delivered from a single institution, limiting the generalizability of the findings. However, pilot studies constitute an important part of the research process [73], and the present findings provide rich data to inform the next steps of PFMT telerehabilitation research. This study also specifically targeted older women with stress or mixed UI, and the results may not be generalizable to other types of UI symptoms or clinical profiles. 

## 5. Conclusions

In conclusion, this study shows that an online group-based PFMT program for older women with UI is feasible from both patient and clinician perspectives. The time constraints due to time spent in individual exchanges emphasized the importance of respecting the program’s pre-established group size of eight women. The study also highlighted the pivotal role of the physiotherapist leading the program in prioritizing the activities according to the clinical objective pursued. Further investigation is needed to determine the clinical effectiveness of the teleGROUP program.

## Figures and Tables

**Figure 1 ijerph-20-05791-f001:**
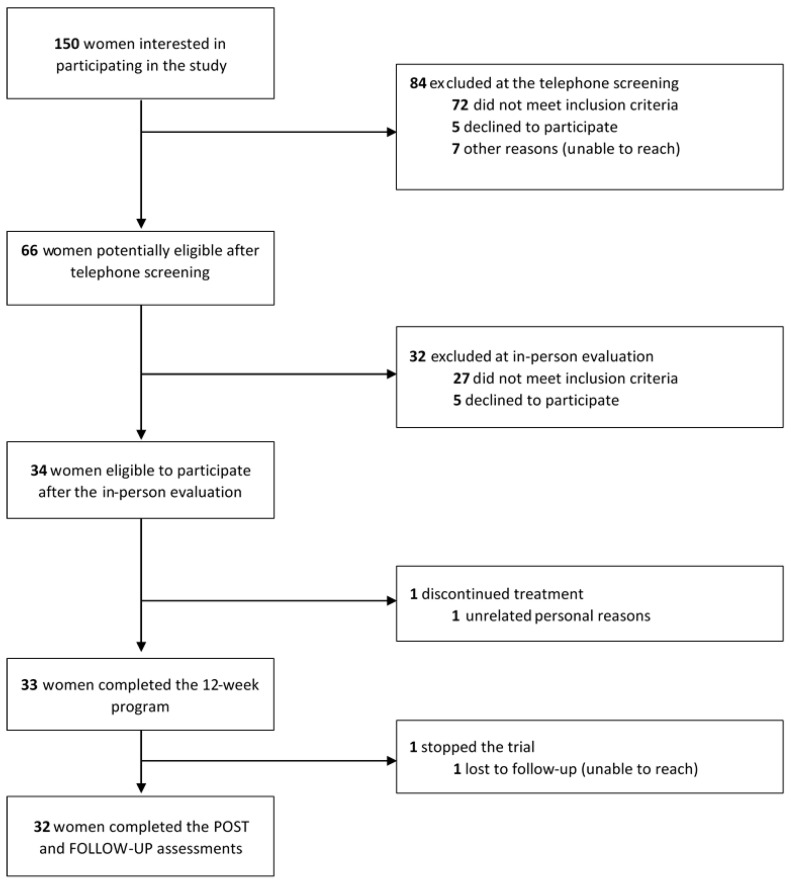
Flowchart of study participants from recruitment and study participation to completion of the final assessments. The bold was used in this figure to highlight the numbers.

**Figure 2 ijerph-20-05791-f002:**
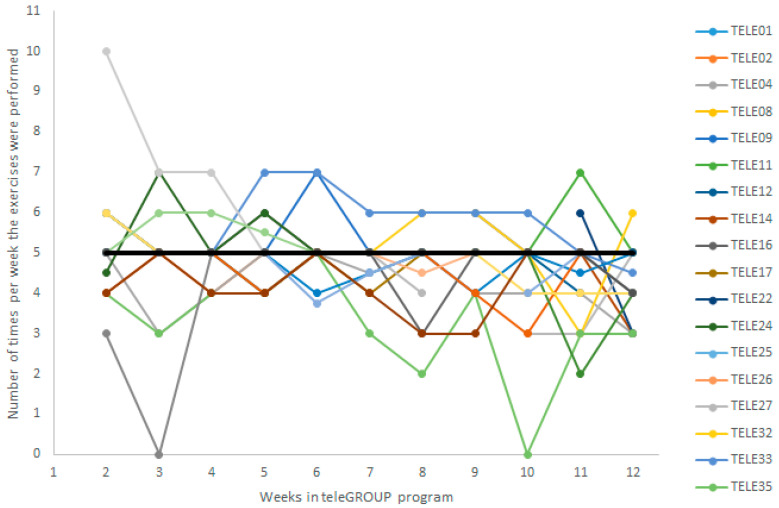
Number of times per week each participant reported performing the exercises throughout the 12-week teleGROUP program. Each participant, labeled with a unique alphanumeric identifier visible on the right, is represented in a different color, and the weekly median is represented in black. As most participants completed their exercises 5 times per week, their data overlap under the median line.

**Table 1 ijerph-20-05791-t001:** Characteristics of participating women (*n* = 34).

Sociodemographic Characteristics	
Age, years (median, IQR)	69.0 (6.0)
Living alone (*n*, %)	13 (38.2)
Civil status (*n*, %)	
Single	6 (17.6)
Married	13 (38.2)
Common-law	8 (23.5)
Divorced	3 (8.8)
Widowed	4 (11.8)
Separated	0 (0)
Education level (*n*, %)	
Elementary	0 (0)
High school	6 (17.6)
College	9 (26.5)
University	19 (55.9)
Annual household income, CAD (mean, SD)	71,909.09 (39,864.90)
General Health Characteristics and Participants’ Symptoms	
Type of urinary incontinence:	
Stress urinary incontinence (*n*, %)	2 (5.9)
Mixed urinary incontinence (*n*, %)	32 (94.1)
Urinary incontinence symptoms duration, years (median, IQR)	5.5 (13.5)
MMSE score ^a,b^/30 (median, IQR)	29.0 (1.0)
MoCA score ^a,c^/30 (median, IQR)	28.0 (3.0)
Number of weekly urine leakages (median, IQR)	13.5 (16.0)
ICIQ-UI ^d^ score/21 (mean, SD)	12.5 (3.0)
Confidence in ability to contract pelvic floor muscles ^e^/100 (median, IQR)	80.0 (20.0)
OTSES score ^e,f^/120 (mean, SD)	60.8 (20.7)
Intravaginal Evaluation	
Observations by the evaluating physiotherapists:	
Perineal inversion (straining and depressing the pelvic floor rather than executing the expected inward lift and squeeze) (*n*, %)	2 (5.9)
Compensations through the contraction of other muscles (i.e., abdominal muscles, gluteal muscles, adductors, diaphragm) (*n*, %)	30 (88.2)
Difficulty relaxing the pelvic floor muscles (*n*, %)	1 (2.9)

^a^ *n* = 27; ^b^ Mini Mental State Examination; ^c^ Montreal Cognitive Assessment; ^d^ International Consultation on Incontinence Questionnaire module on UI symptoms short form; ^e^ *n* = 33; ^f^ Online Technologies Self-Efficacy Scale.

**Table 2 ijerph-20-05791-t002:** Feasibility of the teleGROUP program from the participant perspective (*n* = 34).

Attendance	
Percentage of overall attendance for the scheduled weekly sessions ^a b^	95.2
Number of sessions attended by participating women ^a^ (median, IQR)	12.5 (2.0)
Adherence	
Number of days/week the participants completed the home exercises ^a^:	
Exercise 1: Maximal contractions (median, IQR)	5.0 (0.0)
Exercise 2: Cough (median, IQR)	5.0 (0.0)
Exercise 3: Fast contractions (median, IQR)	5.0 (0.0)
Exercise 4: Podium (median, IQR)	5.0 (0.0)
Overall (median, IQR)	5.0 (0.0)
No. participants who completed their exercises ^a^:	
An overall of 5 days/week or more (%)	21 (63.6)
An overall of 4 days/week (%)	9 (27.3)
An overall of 3 days/week (%)	3 (9.1)
Complications or Side Effects and Dropout	
Participants who reported complications or side effects (*n*, %) ^a^	1 (3.1)
Participants who dropped out during the teleGROUP program (*n*, %)	1 (2.9)
Satisfaction and Experience	
Satisfaction ratings on the single-item question ^c^:	
Completely satisfied (*n*, %)	23 (71.9)
Somewhat satisfied (*n*, %)	8 (25.0)
Not satisfied (*n*, %)	1 (3.1)
Percentage of satisfaction on the visual analog scale ^c^ (median, IQR)	75.0 (30.0)
SUS score ^c,d^/100 (median, IQR)	93.8 (15.0)

^a^ *n* = 33; ^b^ Attendance was calculated for the 12 scheduled weekly sessions and did not consider the thirteenth optional session offered. ^c^ *n* = 32; ^d^ System Usability Scale.

**Table 3 ijerph-20-05791-t003:** Questionnaire scores of the physiotherapists conducting the initial individual in-person evaluation session (based on the Theoretical Framework of Acceptability) [58] (*n* = 14).

Domains	Questions Rated from 0 to 10, 0 Being ‘Not at All’ and 10 Being ‘Completely’	Scores/10 (Median, IQR)
Ethicality	To what extent did the initial individual in-person evaluation have a good fit with your personal values?	10.0 (9.0–10.0) ^a^
Affective attitude	To what extent did you enjoy conducting the initial individual in-person evaluation and being part of the program?	10.0 (9.0–10.0)
Burden	To what extent did you find the effort required to complete the training, schedule the appointment using the participant’s contact information and conduct the assessment reasonable?	9.0 (7.8–10.0)
Opportunity costs	To what extent were you able to maintain your benefits, profits or advantages while conducting the evaluation? (e.g., no associated loss of business, or fear of loss of business)	10.0 (8.0–10.0)
Opportunity costs	To what extent were you comfortable with the online classes being offered by an organization external to your clinic?	10.0 (9.0–10.0)
Perceived effectiveness	To what extent were you confident that the initial individual in-person evaluation allowed you to determine if the participant was a good candidate for online classes?	9.0 (9.0–10.0) ^a^
Self-efficacy	To what extent were you confident that you would be able to complete the evaluation as planned by the research project?	9.5 (8.8–10.0)
Intervention coherence	To what extent do you feel that the evaluation was consistent and relevant before participants took part in the teleGROUP program?	10.0 (8.0–10.0)

^a^ *n* = 13.

**Table 4 ijerph-20-05791-t004:** Conformity to the original program guidelines.

	Completed per Protocol (%)	Program Adaptations		Not Completed (%)
		Completed in Another Position (%)	Additional Sets Completed (%)	
Non-exercise Components				
Individual exchanges with the physiotherapist	100	0.0	0.0	0.0
Education session on pelvic floor-related topics and motivational capsules	100	0.0	0.0	0.0
Warm-up and Flexibility Movements				
Anterior pelvic tilt	100	0.0	0.0	0.0
Lateral pelvic tilt	87.5	0.0	12.5	0.0
Simple pelvic rotation	85.4	0.0	14.5	0.0
8-figure pelvic rotation	87.5	0.0	12.4	0.0
Ankle movements	8.3	0.0	0.0	91.7
Pelvic Floor Muscle Exercises				
Maximal pelvic floor contraction exercise	64.6	31.2	0.0	3.3
Knack (cough) exercise	62.9	33.3	0.0	3.0
Podium contraction exercise	42.9	45.2	10.3	1.6
Fast contractions	30.6	26.4	42.8	0.0
Moderate sustained contraction	43.8	18.8	0.0	37.5
Functional exercise: Dance activity	31.2	0.0	0.0	68.7
Additional Exercises				
Deep breathing	0.0	0.0	0.0	100.0
Transverse abdominal contraction	12.5	3.1	0.0	84.4

## Data Availability

The data presented in this study are available upon request from the corresponding author.

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
