# Peer review of "Group-Based Pelvic Floor Telerehabilitation to Treat Urinary Incontinence in Older Women: A Feasibility Study"

_ijerph, 2023, doi:10.3390/ijerph20105791_

Round 1

Reviewer 1 Report

The abstract needs to have an introduction about UI. I am unsure about the rationale of the study. Why is it even important to pursue this study?

I am unsure of the novelty of this and the overall purpose, significance, and impact.

Also, the manuscript requires English language copyediting.

The introduction is also very short. It does not also align with the abstract.

The title is somewhat odd. The first sentence ends with a question mark but then it was followed by a phrase 'a feasibility study'. I think revising this is needed.

I was not able to completely comprehend the manuscript and what the overall goal is because the introduction nor the abstract was not clear.

What previous studies were done? What is the definition of a feasbility study?

Reviewer 2 Report

In view of a feasibility study, in the introduction it is not clear whether women do not receive treatment because they do not seek help and, in this case, why. Or, They seek help and there is no response capacity. when carrying out a feasibility study, the project methodology must be clear, for example SWOT/FOFA analysis; risk assessment or other . Are we dealing with action research or is it really a feasibility study? Are we studying the possibility of finding an alternative, or the effectiveness of intervention?

Reviewer 3 Report

The authors describe the feasibility of a telerehabilitation program for treating urinary incontinency. Attrition rate showed that at least 15 women (10 not interested, others discontinued) did not show interest in the program. This compromises feasibility. In this line, it is somehow surprising that the recruitment trial is just exactly as anticipated (n=32, which is the exact number of women indicated in reference number 23). Finally, feasibility studies must also inform about the economic cost of its development.

Under these circumstances, my advice is to show a complete study with a double fold objective. Firs, feasibility. Second, effects of the program on UI.

Reviewer 4 Report

Thank you for this interesting research. I am very satisfied with the study. I  want to highlight some topis to improve: 

- Lines 69 and 91: " available elsewhere"?

- Inclusion criteria: in some patients with UI , pelvic floor strength exercises are not recommended. If physiotherapists have considered this issue in the evaluation visit, it should be mentioned. 

- Participants : We have the pre bladder diary of all of them so it would be : good to have information about the leakage evolution after the program: Have the number of urinary leakage decreased? How?

- Why chronic constipation is a clinical profile incompatible with the study? Explain.

- Was only a F2F session enough to be sure participants can contract their pelvic floor properly? Explain

- Line 304. Clarify, sentence confuse. 

- The PFM program shoud be better described: type of excerside, lenght of contraction, number of repetitions, etc, to assure the study protocol could be replicated. 

Round 2

Reviewer 3 Report

In spite of the changes made, I still think that threre are some methodological issues that can not be solved and jeopardize the quality of the work.
